# Targeting the Tetraspanins with Monoclonal Antibodies in Oncology: Focus on Tspan8/Co-029

**DOI:** 10.3390/cancers11020179

**Published:** 2019-02-03

**Authors:** Mathilde Bonnet, Aurélie Maisonial-Besset, Yingying Zhu, Tiffany Witkowski, Gwenaëlle Roche, Claude Boucheix, Céline Greco, Françoise Degoul

**Affiliations:** 1Université Clermont Auvergne, INSERM1071, Microbes, Intestins, Inflammation et Susceptibilité de l’hôte, 63001 Clermont-Ferrand CEDEX 1, France; mathilde.bonnet@uca.fr (M.B.); gwenaelle.roche@uca.fr (G.R.); 2Université Clermont Auvergne, INSERM U1240, Imagerie Moléculaire et Stratégies Théranostiques, F-63000 Clermont Ferrand, France; aurelie.maisonial@uca.fr (A.M.-B.); tiffany.witkowski@inserm.fr (T.W.); 3Université Paris-Sud, INSERM U935, Bâtiment Lavoisier, 14 Avenue Paul-Vaillant-Couturier, F-94800 Villejuif, France; julianzyy@hotmail.com (Y.Z.); claude.boucheix@inserm.fr (C.B.); celine.greco@inserm.fr (C.G.); 4Department of Pain and Palliative Medicine AP HP, Hôpital Necker, 75015 Paris, France

**Keywords:** tetraspanins, cancer, Tspan8, immunotherapy, radioimmunotherapy

## Abstract

Tetraspanins are exposed at the surface of cellular membranes, which allows for the fixation of cognate antibodies. Developing specific antibodies in conjunction with genetic data would largely contribute to deciphering their biological behavior. In this short review, we summarize the main functions of Tspan8/Co-029 and its role in the biology of tumor cells. Based on data collected from recently reported studies, the possibilities of using antibodies to target Tspan8 in immunotherapy or radioimmunotherapy approaches are also discussed.

## 1. Introduction

Tspan8 belongs to the tetraspanin molecular family of surface glycoproteins containing 33 members in humans, which are now referred to as Tspan1–33 (Figure 1). Tetraspanins are small membrane proteins (200–350 amino acids), which interact laterally with multiple partner proteins and with each other to form the so-called TEMs (tetraspanin-enriched microdomains). To qualify for membership in the tetraspanin family, a protein must have four transmembrane domains and several conserved amino acids, including an absolutely conserved CCG motif and two other cysteine residues that contribute to two crucial disulfide bonds within the second extracellular loop (EC2) (Figure 2). One or two additional disulfide bonds may also be found in EC2. The biological importance of tetraspanins is supported by functional consequences of genetic modifications that occur either spontaneously in humans or experimentally in mice. For instance, gene inactivation may affect fertility (CD9, CD81), and visual (RDS), kidney (CD151) or immunological functions (CD81, CD37) [1,2,3,4]. Since tetraspanins are not adhesion or signaling molecules, receptors or enzymes, the properties of these molecules are highly dependent on their ability to form TEMs with a hierarchical organization. Indeed, each tetraspanin has specific partners, including integrins, ADAM metalloproteases, growth factor receptors and histocompatibility antigens, that they are directly associated with through protein–protein interactions and form primary complexes with [5,6]. Coupled together, the latter can form second-order complexes through tetraspanin–tetraspanin interactions that may involve cholesterol and palmitoylation. In some cases, the function of these associated molecules has not yet been elucidated for the CD9P-1 and EWI2 (official protein names of PTGFRN and IGSF8) [7,8]. Some molecular experiments showed that tetraspanins may interfere with the properties (affinity of the integrin α6β1 for laminin-1 is modulated by the tetraspanin CD151) [9], the trafficking (CD81 controls the expression of CD19 at the B-lymphoid cell surface) [10] or membrane compartmentalization of associated molecules (such as ADAM10 by TspanC8) [11]. Detailed proteomic analysis of membrane molecules that are able to associate with CD9 and Tspan8 has been reported recently in colon carcinoma cell lines. Among other membrane proteins, E-cadherin and EGFR are associated with TM4 complexes. More specifically, the presence of Tspan8 in the membrane drives EGFR to tetraspanin complexes, which results in changes in motility behavior (see next paragraph) [12].

Some tetraspanins are widely expressed (for example, CD81 and CD151) while others are more restricted (for example, CD37 and CD53 on lymphoid cells or RDS in retina). Several clinical observations and experimental data have correlated the expression of some tetraspanins with tumor metastatic potential. CD82 or CD9 expression is generally associated with favorable prognosis in different cancers [3]. On the contrary, CD151 and Tspan8 expression in tumor cells has been frequently associated with increased migration, proliferation and angiogenesis induction (Table 1). Mechanisms that could potentially explain the role of tetraspanins in tumors have been investigated experimentally using cellular models of overexpression in addition to knockdown or knockout animal models (reviewed in references [3,4,14,15,16]). The modulation of cellular properties, such as proliferation, cell migration and apoptosis, has been previously reported. Among other mechanisms, the knockout of CD37 leads to the occurrence of lymphomas that appear to be linked to the constitutive activation of the IL6 signaling pathway [17]. At a tissue level, an effect of Tspan8 on angiogenesis could be partially mediated by exosomes. More generally, the role of tetraspanins in tumor cell communication with their microenvironment through an effect on exosomal biogenesis is considered to be an important function of these molecules [18]. The changes in cell properties induced by the (over)expression of Tspan8 have been investigated in preclinical models. For example, the Isreco1 cell line derived from primary colorectal cancer (CRC) that does not express Tspan8 was compared with Is1-Co029, which was obtained by transduction to express Tspan8 at the same level as the two cell lines derived from metastases of the same patient. There was no difference in the motility of single cells plated on collagen, but RNAi targeting various surface molecules, such as E-cadherin, p120-catenin or EGFR, increased the motility of Is1-Co029, whereas no effect was observed on Isreco1 [12,19]. A possible link between Tspan8, E-cadherin and motility could be the signaling molecule p120-catenin, which is retained at the cell membrane through its affinity for E-cadherin. Furthermore, it has been reported to regulate Rho and Rac functions in cell adhesion and motility (reviewed in reference [19]). However, models using long-term established tumor cell lines in 2D settings could be misleading and it would be important to conduct experiments on cells derived from fresh tumors and cultured in 3D conditions as organoids that would better reflect the in-vivo conditions.

Experimental results and clinical trials using monoclonal antibodies (mAbs) suggest that some tetraspanins may be targeted in hematological malignancies and carcinomas [26]. In a similar way, Tspan8 appears to be an interesting candidate for the development of therapeutic antibodies, and different methods are discussed in the last part of this review.

## 2. Mechanisms and Requirements for the Use of Therapeutic Anti-Tetraspanin Antibodies

### 2.1. Molecular Mechanisms

Since antibodies are large proteins, their ability to reach their target may be limited by their diffusion inside different tissues and their components due to certain barriers, such as the hemato-encephalic barrier. However, since the permeability of tumor vessels is usually abnormal, diffusion appears to be increased in such tumor tissues [27,28]. Once located close to their biological target, the antibodies can enact different mechanisms of action.

#### 2.1.1. Blocking Antibodies

According to current knowledge, the action of tetraspanins relies on their ability to regulate the function of their partner molecules. Even if the detailed molecular basis remains mostly unknown, the binding of mAbs on Tspan8 may result in the inhibition of cell migration, invasion, proliferation and angiogenesis in organized tissues (Table 1).

#### 2.1.2. Cytotoxic Antibodies

Unconjugated antibodies mediate ADCC (antibody-dependent cell cytotoxicity) via the activation of accessory cytotoxic cells for killing target cells. Upon fixation on the cell surface, they recruit macrophages or NK cells through their Fc fragment, which can further destroy the target cell. The subclass and glycosylation of the Fc region are important parameters that determine the efficacy of ADCC. Furthermore, the ability of antibodies to mediate ADCC may be optimized through genetic modification of their Fc region.

On the contrary, radionuclide- or drug-conjugated antibodies have directly toxic effects in vivo.

Depending on the radionuclide used (nature of emissions, energy and so on), radionuclide-conjugated antibodies can efficiently treat the target cells and the neighboring cells, which can be of high value in the case of heterogeneous cell expression of the target molecule. In addition, radionuclide-conjugated antibodies can also allow for a combination of imaging and radiotoxicity. This will be detailed in the radionuclide-conjugated Tspan8 antibodies section.

When antibodies are conjugated with cytotoxic drugs, an internalization of the antigen/antibody complex is often necessary to allow drug delivery [29]. However, if the targeted protein is widely distributed, the use of these two main categories of antibody–drug conjugates may have deleterious effects on normal tissues. As a consequence, the careful evaluation of the pattern of expression of the target is a critical prerequisite for the development of a therapeutic antibody. However, recent progress in antibody-conjugated drugs has allowed one to target more specific released molecules in tumors. For instance, cleavage in acidic zones that occur in the highly proliferative zone or proteolytic cleavage by tumor enzymes, such as MMPs, increase specificity. Another stage was added with a pro-body approach, which consists of modifying the antibody with a small peptide that needs to be cleaved by a specific protease for antigen recognition [30].

### 2.2. Pattern of Expression

An ideal antigenic target should be expressed at a high intensity on the surface of tumor cells, especially tumor stem cells and not on normal cells. None of the molecules that are currently targeted by mAbs are able to fulfil these criteria and not surprisingly, tetraspanins do not escape this rule.

As mentioned above, the tissue distribution of tetraspanins is highly variable. A very restricted distribution of some tetraspanins has already been observed for peripherin/RDS found in the photoreceptors, UPk1a found on the urinary bladder epithelium or CD37 expressed mainly on B lymphoid cells. In contrast, other tetraspanins, such as CD9, CD63, CD81 or CD151, have a very large distribution and may be difficult to target in vivo. Tspan8 is expressed in a limited number of tissues and is mainly found on epithelial cells of the digestive tract. However, it is also located on the epithelial cells of the kidney, prostate and trachea [23]. The second aspect relates to the intensity ratio of Tspan8 in tumor compared to normal tissues, which is another crucial issue that needs to be addressed in order to avoid side effects. Increased Tspan8 expression in tumor tissues (colorectal and ovarian cancer, melanoma, hepatocellular and pancreatic carcinoma) is generally related to worse prognosis [19,25,31,32,33]. Therefore, Tspan8 appears to be an interesting target candidate for cancer treatment with mAbs.

### 2.3. Dodging Immune Neutralization

This important point has been widely studied and previously reviewed. Thus, this will not be detailed in this article. This is mainly realized through the humanization of the molecule but other methods have been previously reported [34].

## 3. Therapeutic Antibodies Directed toward the Tetraspanins CD9, CD151 and CD37 in Cancer

There are a few proofs of principle that have shown that targeting tetraspanins with antibodies might inhibit tumor growth or even induce partial or complete remission [35].

The effects of the anti-CD9 mAb ALB6 (IgG1) injected intravenously were investigated in a model of human gastric cancer (MKN-28) implanted subcutaneously in nude mice. A reduction of 60–70% in the size of the tumor in the treated group was observed compared to the control IgG-treated mice. At the same time, a significant reduction in cell proliferation and angiogenesis and an increase in apoptotic signals were observed [36]. Since this mAb is directed towards human CD9, damages to normal tissues were not evaluated. As CD9 is strongly expressed on many cellular types and particularly on platelets in humans, the use of this anti-CD9 antibody could lead to a loss of treatment efficiency and may trigger platelet activation or lysis depending on the nature of the Fc fragment [37]. However, Fc-mediated side effects could be avoided by genetic modifications of the antibody.

There have only been a few studies targeting tetraspanins in humans for therapeutic purposes. A cocktail of antibodies Ba1/2/3 directed respectively toward CD9, CD24 and CD10 to deplete the bone marrow of acute lymphoblastic leukemia patients for autologous bone marrow transplantation was investigated. Even if the specific role of each individual antibody was difficult to evaluate, an absence of toxicity against hematological stem cells was observed in these antibodies [38].

Due to its restricted specificity for differentiated cells of the B-cell lineage, CD37 was considered to be a potential target for the treatment for B-cell lymphoma using an anti-CD37 antibody radiolabeled with iodine-131 (β^−^; T_1/2_ = 8.02 d; 606 keV). Very encouraging results were obtained in comparison with an anti-CD20 antibody labelled with the same radionuclide [39]. However, since the use of unconjugated humanized anti-CD20 mAbs for B-cell lymphoma has been a very straightforward method for the improvement of treatment protocols, the use of anti-CD37 mAbs for the treatment of lymphoma and chronic lymphocytic leukemia was abandoned and only recently reintroduced. Several forms of therapeutic anti-CD37 antibodies, whether they are human or humanized, used in combination or alone (unconjugated antibodies (Bi 836826 and otlertuzumab) and drug conjugates (monomethyl auristatin E: AGS67E, maytansine: IMGN529)), or radiolabeled antibodies (^177^Lu: belatutin) [40], have been developed and are currently undergoing clinical tests. Promising results have been obtained in the resistant forms of NHL (Non-Hodgkin Lymphomas) and CLL (Chronic Lymphocytic Leukemias).

Anti-CD151 antibodies inhibited metastasis spread and primary tumor growth in human tumor mouse models [41,42]. Despite their variable ability to disrupt the complex between CD151 and α3β1 integrin, different anti-CD151 antibodies were reported to prevent metastasis formation in clinical models [35].

## 4. Treatment with Unconjugated Anti-Tspan8 Antibodies

Unconjugated mAbs may act through two different mechanisms, which are namely the mediation of ADCC or interference with molecular functions that are required for malignant cells to express their tumorigenicity. The functional activity of the anti-Tspan8 mAbs can be assessed to some extent in vitro but the link between tumor growth inhibition and either ADCC or functional inhibition could be difficult to determine in vivo.

Several studies were performed by Zöller’s group with the rat pancreatic adenocarcinoma cells (AS-Tspan8 compared to AS). The anti-rat Tspan8 mAb D6.1 was found to inhibit cell proliferation in vitro [20]. Furthermore, in rat mesentery fragments cultured with tumor cells or their exosomes, increased endothelial cell branching linked to Tspan8 expression was blocked by mAb D6.1 [21]. These studies suggest that Tspan8 overexpression in rats promotes angiogenic activity and supports tumor growth while anti-rat Tspan8 mAbs can efficiently inhibit this process. In addition, increased vessel density observed by intravital microscopy was abolished after treatment with D6.1 in an in-vivo model of peritoneal carcinosis [21].

Anti-human Tspan8 mAbs (Ts29.1: IgG1 and Ts29.2: IgG2b) were produced by Boucheix’s team. These antibodies had no effect on cell proliferation, migration or apoptosis for colorectal cancer (CRC) cell lines Isreco-1/Is1-Co029, SW480 (presenting spontaneously weak Tspan8 expression) and SW480-Co029 (overexpressing Tspan8 after gene transduction) in vitro. However, the 2D motility of Is1-Co029 was increased by RNAi targeting E-cadherin and p120-catenin while it only decreased after specific co-treatment with anti-Tspan8 mAbs [12,19]. EGFR blocking (mAb cetuximab or AG1478, a chemical EGFR inhibitor) in Is1-Co029 cells also induced an increase in cell motility, which was further blocked by treatment with anti-Tspan8 mAbs [12]. The observations in relation to EGFR inhibition were unexpected since Isreco-1 cell lines have a KRAS mutation, which should be associated with an inhibition of the EGFR function. Thus, this suggested that EGFR signaling may still be influenced when Tspan8 is expressed.

In a mouse model of CRC (SW480 vs. SW480-Co029), the growth of SW480-Co029 tumors was inhibited by up to 70% when treated in the early stages with the IgG2b anti-human Tspan8 mAbTs29.2 in vivo (initially 2 mg intraperitoneally, followed by 1 mg twice a week for 4 weeks). The same results were also observed in another CRC mouse model, which expressed spontaneously high levels of Tspan8 (HT29). The inhibition of the cell proliferation in vivo was demonstrated by a reduction of the mitotic index in HT29 tumor cells in Ts29.2-treated mice. These in-vivo data underlined the crucial role of Tspan8 in tumor growth and the therapeutic potential of anti-Tspan8 mAbs as a CRC treatment. The discrepancy between the in-vitro and in-vivo data on cell proliferation suggested that the binding of Ts29.2 to tumor cells may modify their response to signaling from the microenvironment. No significant differences between the treated and control mice were found when assessing the inflammatory infiltrate, angiogenesis (CD34) and apoptotic signal (Casp3). These findings did not support the hypothesis of ADCC.

In another approach targeting Tspan8, Park et al. [25] used phage display technology to produce a fully human mAb directed against Tspan8 LEL (large extra loop = EC2). For an in-vivo experiment, the mice that were intraperitoneally injected with SK-OV3-luc human ovarian cell line intravenously received either IgG or Tspan8–LEL IgG (10 mg/kg) twice a week until day 42 post inoculation. In the control IgG-treated group, a detectable luminescence signal in removed organs (ovary, pancreas, colon, heart, liver, spleen and kidney) was observed in 24 of 31 control mice whereas the incidence fell to 50% (15 of 30 mice) in the Tspan8–LEL IgG-treated group. This reduction of 35% was considered to be significant and the mice did not show any signs of severe toxicity.

## 5. Treatment with Radionuclide-Conjugated Anti-Tspan8 Antibodies

### 5.1. Radiolabeling of Antibodies

For radiotherapeutic purposes, antibodies are usually modified with grafted chelating moieties to allow radiolabeling with β^−^-emitter radionuclides like yttrium-90 (β^−^; T_1/2_ = 2.67 d; 2280 keV) or lutetium-177 (β^−^; T_1/2_ = 6.65 d; 498 keV) [43]. Advantageously, lutetium-177 can be used for both imaging and therapeutic purposes as β^−^ and γ radiations are generated during its decay. To date, the only radiolabeled mAb authorized for human treatment is the anti-CD20 [^90^Y]ibritumomab tiuxetan (Zevalin^®^), which is administered as a second-line treatment to patients with non-Hodgkin lymphomas (NHLs) that are resistant to chemotherapy [44]. Although they were created several decades ago, radioimmunotherapy (RIT)sing β^−^-emitter—conjugated mAbs remains underused due to inherent issues concerning hematotoxicity, which is induced by the long biological half-life of mAbs in blood, and the low penetration of antibodies in solid tumors [43,45]. Thus, different strategies have been proposed to enhance RIT efficiency, including the use of α-particle emitters, pretargeting protocols and reduction/modification of the antibody size (nanobodies, affibodies and so on).

The use of α-particle emitters is of great interest for delivering high linear energy transfer (LET) in very small volumes (cell diameters of 50–100 µm) without affecting neighboring tissues. Among the increasing number of clinical studies using mAbs/peptides/ligands/radionuclides delivering α-particles [46], two mAbs radiolabeled with bismuth-213 (α; T_1/2_ = 45.6 min; 5869 keV) (^213^Bi-cDTPA-9.2.27 targeting MSCP in melanoma and ^213^Bi-Hum195 mAb targeting CD33 in acute myeloid leukemia) have had positive results in terms of prognosis [47,48]. Moreover, challenging approaches using DOTA-mAbs radiolabeled with actinium-225 (α; T_1/2_ = 10.0 d; 5580–5830 keV) are currently under investigation as this radionuclide decay generates francium-221 (α; T_1/2_ = 4.79 min, 6300 keV), which has an interesting secondary radiotoxic effect [49]. This strategy should be applied for internalizing antibodies to concentrate α-particles in tumor cells.

To decrease hematotoxicity, different pretargeting strategies were developed to dissociate the mAb antibody and radionuclide injections [50]. One of the most interesting approaches involves the use of bio-orthogonal chemistry. In such strategies, the mAb and the delayed injected radioactive molecule are grafted with chemical entities that are highly reactive to each other but inert to chemical functions usually found on in-vivo molecules, such as proteins. This fast and specific reaction can advantageously take place in aqueous media, which is compatible with in-vivo applications. Another way of reducing side effects of RIT is to inject the radioactive molecule in a specific organ to avoid systemic irradiation. For example, this might involve intrahepatic metastases using arterial infusions [51].

Finally, different small protein forms (affibodies, nanobodies and so on) ranging from a few to 30 kDa with a shorter biological half-life in the blood circulation have been tested in preclinical RIT protocols [52] but mostly in nonradioactive applications. In clinical trials, one affibody is currently being investigated in Her2 breast cancer imaging [53].

### 5.2. Targeting Tspan8 with Radiolabeled Antibodies

Our team investigated the biodistribution of two monoclonal antibodies targeting human Tspan8, which were namely Ts29.1 and Ts29.2. These were grafted with DOTA and radiolabeled with indium-111 (γ; T_1/2_ = 2.80 d; 171 keV; 245 keV). The measurement of the immunoreactive fraction revealed that the addition of DOTA-chelating moieties and radiolabeling did not modify the affinity of Ts29.2 for its target. The uptake of [^111^In]DOTA-Ts29.2 in HT29 tumors was higher than that of [^111^In]DOTA-Ts29.1 (Figure 3A,B). After this, further experiments were conducted on different models of xenografts [23]. Biodistribution studies on mice with both SW480-Co29/SW480 tumors demonstrated high specificity of [^111^In]DOTA-Ts29.2 for Tspan8-expressing tumors. The same results were obtained using the Isreco-1 and Is1-Co029 models (Figure 4A,B). Further RIT experiments using this antibody were supported by the promising biodistribution and dosimetry results collected. During therapeutic studies, we observed that [^177^Lu]DOTA-Ts29.2 induced a significant reduction in HT29 xenograft growth, with molecular events sustaining the effects of the radiation in this model.

To initiate the pretargeting strategies, Ts29.2 was also modified by the addition of a transcyclooctene (TCO) to the lysine residues, which was evaluated in studies conducted in vitro and in vivo using a fluorescent tetrazine. We evaluated the best link size between TCO and TS29.2 and observed a higher fluorescent signal with Ts29.2-TCO without a PEG spacer, which can be explained by a higher isomerization rate of TCO to the inactive CCO form [54]. As tetrazin can be conjugated to a DOTA group, RIT with β^−^-emitters or α-particles will be considered. A recent preclinical study using such an approach had significant effects on mice xenografted with ovarian tumors and treated with an anti-CEA-TCO for 72 h before radionuclide injection [55].

### 5.3. Pros and Cons of RIT for Human Cancers: Focus on Targeting Tspan8

Stoichiometrically compared to its corresponding nonradiolabeled antibodies, [^177^Lu]DOTA-Ts29.2 induced a greater slowing down of tumor growth. The main features in pretargeted radioimmunotherapy PRIT experiments were the reduction of proliferation and increase in apoptosis. As mentioned above, the treatment with nonradioactive antibodies (using 100-times more antibodies than in the [^177^Lu]DOTA Ts29.2 experiments) also resulted in a slowing down of tumor growth with neither induction of apoptosis nor decrease in angiogenesis. In fact, the nonradioactive antibody should alter the interactions between tumor cells harboring Tspan8 and the microenvironment while its radiolabeled counterpart irradiates all surrounding cells after it attaches to its target antigen. This property should be interesting as it will decrease the number of so-called cancerous stem cells (CSCs) because Tspan8 has been identified on the surface of CSCs in pancreatic tumors [56]. RIT has been proven to be effective in stopping CSCs in melanomas using preclinical models, which utilized an IgM directed toward melanin and radiolabeled with rhenium-188 [57]. Conversely, Tspan8 is exposed on the surface of circulating exosomes [22], leading to potential blood radiotoxicity in RIT experiments. Apart from this potential disadvantage, one can imagine that targeting circulating exosomes will be of interest as these vesicles are implicated in metastatic spread [58]. As mentioned above, the hematotoxicity might be prevented by pretargeting strategies, which will be further reinforced by the use of blood clearing agents such as nonradiolabeled ligands conjugated to albumin [59]. As an example, this might allow their metabolism in the liver.

Tspan8 expression is restricted and this protein has been described as a significant contributor and potential therapeutic target in several cancer types. Even if secondary effects and immune system involvement cannot be evaluated on tumor-grafted mouse models used for these studies, targeting Tspan8 with radiolabeled antibodies seems to be an effective antitumoral therapy.

## 6. Conclusions

Tetraspanins may have a broad range of actions in cancers due to their intrinsic membrane localization (cell membrane or exosomes) and high numbers of their interacting molecules [3,26]. The aim of this article was to review recent preclinical attempts at targeting tetraspanins in cancer with a focus on Tspan8. Unconjugated antibodies and radionuclide-conjugated antibodies conceptually represent two different approaches for killing cancer cells through the expression of a surface molecule. Antibodies may have complex effects as they combine cell-mediated cytotoxicity and functional deleterious effects, such as apoptosis induction, or invasive growth and angiogenesis inhibition. This can occur directly or through microenvironment factors. For tetraspanins, it is still unknown how the targeting can alter the function of tumor cells in vivo, but their association with adhesion molecules, growth factor receptors or enzymes inside membrane molecular complexes leads to disturbance of the structure/composition of these complexes, which may result in modulation of migration and abnormal signaling into the cell and finally, inhibition of invasion/metastasis or even apoptosis. A better view and understanding of the behavior of tumor cells in real life would require improved models (such as 3D in-vitro setups with microenvironment reconstitution or syngeneic models in vivo). Although the mechanism of action of radionuclide antibodies is simple and straightforward, their manufacturing requires careful technical management and radioprotection protocols at all stages of their manipulation. However, these offer interesting potential and should be pursued in the future. Innovative techniques have also been developed to reduce harmful effects that are linked to the antibodies binding to normal healthy tissue.

## Figures and Tables

**Figure 1 cancers-11-00179-f001:**
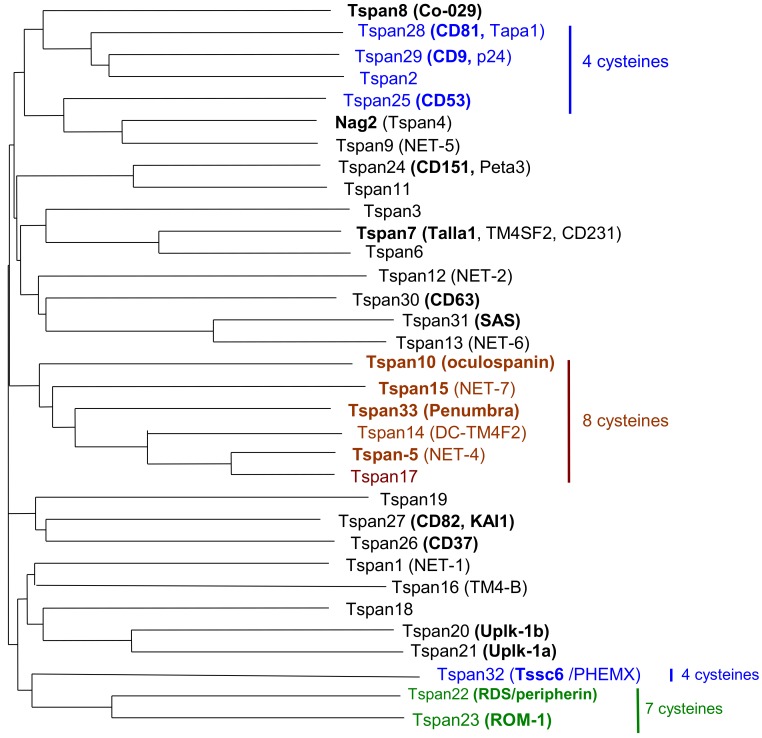
Homology tree between human tetraspanins. Protein sequences have been aligned to generate this distance tree. Bolded names correspond to commonly used ones. The number of cysteines in the extracellular loop EC2 is given when they are different from six cysteines.

**Figure 2 cancers-11-00179-f002:**
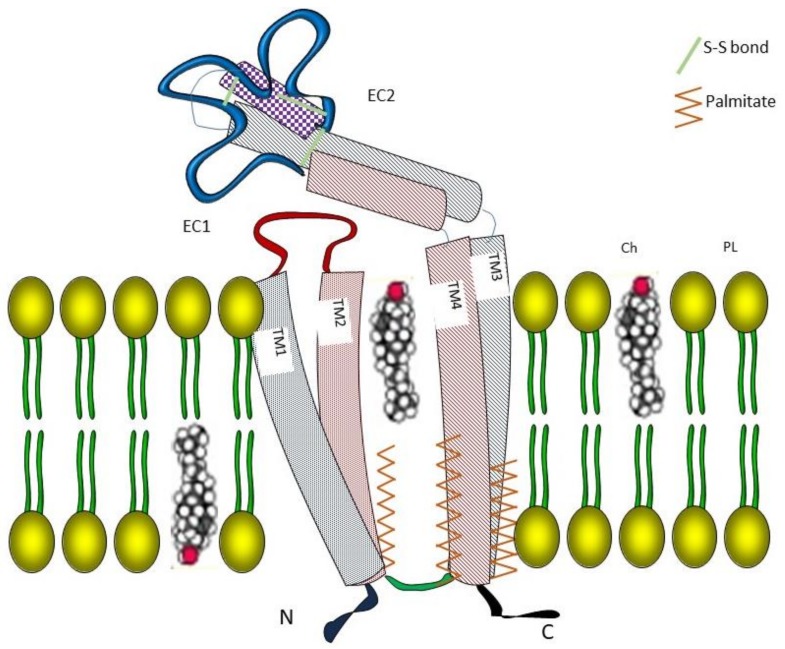
Schematic representation of Tspan8 in membrane. The transmembrane (TM) domains inserted in the phospholipid bilayer are palmitoylated with a cholesterol molecule in between these domains (by analogy with CD81 structure [13]). Among the two extracellular (EC) regions, the larger region contains six cysteines involved in disulfide bonds. EC1 and EC2 = extracellular domains 1 and 2; Ch = cholesterol; and PL = phospholipid.

**Figure 3 cancers-11-00179-f003:**
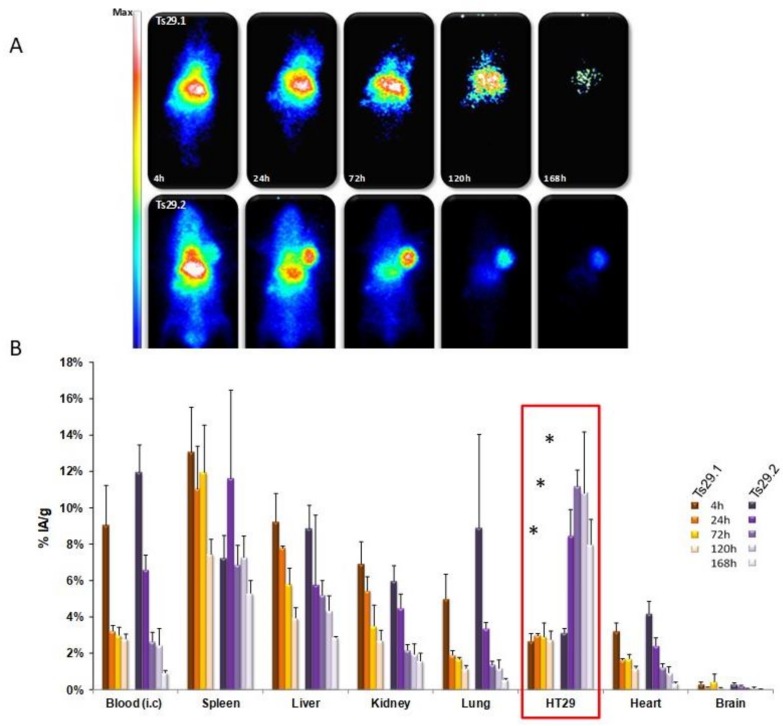
In-vivo selection of [^111^In]DOTA-Ts29 antibodies for imaging and therapy in mice with HT29 colon carcinoma xenografts. Nude NMRI mice with HT-29 tumors were injected (i.v.) with 3.7 MBq of [^111^In]DOTA-Ts29.1 (upper images) or [^111^In]DOTA-Ts29.2 (lower images), which were imaged with a planar γ-camera at 4 h, 24 h, 72 h, 120 h and 168 h post injection (**A**). Biodistribution was performed after euthanasia and expressed as the percentage of activity injected per gram of tissue (%AI/g) of [^111^In]DOTA-Ts29.1 (orange) or [^111^In]DOTA-Ts29.2 (purple) (**B**). Radioactivity was measured using a γ-counter. Results are presented as the average percentage of injected dose/gram of tissue of three animals for each time point. The error bars represent the standard deviation. Biodistribution difference between the two mAbs: * *p* < 0.05 Fisher test.

**Figure 4 cancers-11-00179-f004:**
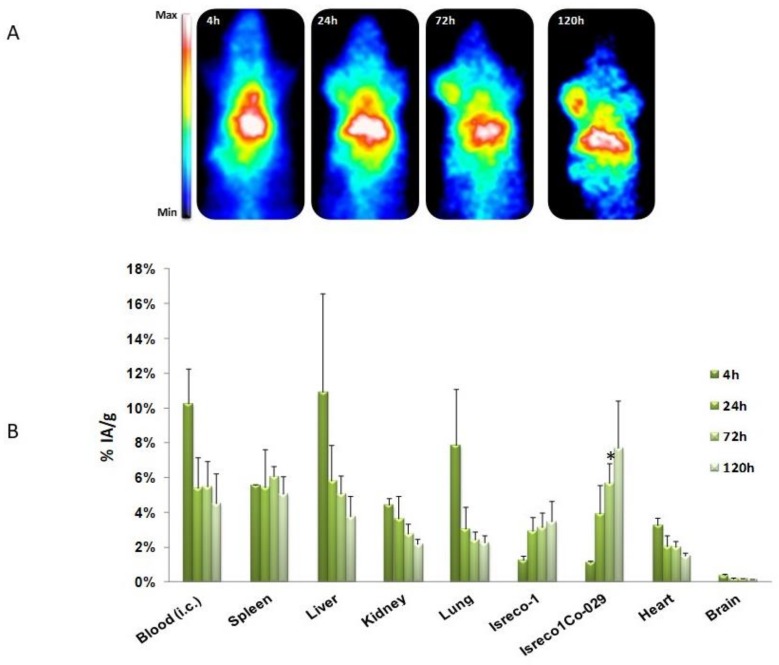
In-vivo specificity of [^111^In]DOTA-Ts29.2 for Tspan8-expressing tumors. Nude NMRI mice with Isreco-1 (left shoulder) and Is1-Co29 (right shoulder) were injected (i.v.) with 3.7 MBq of [^111^In]DOTA-Ts29.2 and imaged with a planar γ-camera at 4 h, 24 h, 72 h and 120 h post injection (**A**). Ex-vivo biodistribution study (%AI/g) of [^111^In]DOTA-Ts29.2 (**B**) was determined on the same mice with the same protocol as Figure 3B. Biodistribution difference between the two tumors: * *p* < 0.05. Fisher test.

**Table 1 cancers-11-00179-t001:** Summary of studies dealing with biological consequences of Tspan8 targeting using specific antibodies.

Preclinical and clinical models	Effect of Tspan8 Expression and Modulation by Antibody or Tspan8-LEL Targeting on Migration/Invasion/Metastasis, Proliferation/Tumor Growth, Angiogenesis	References
Rat pancreatic adenocarcinoma cells (AS-Tspan8 vs. AS)	In vitro: similar proliferation of two AS cell lines Inhibition by anti-rat Tspan8 mAb D6.1 of AS-Tspan8 In vivo: increased metastasis formation of AS-Tspan8 (i.v., s.c. or i.f.p. injection)	[20]
Rat pancreatic adenocarcinoma cells (AS-Tspan8 vs. AS)	In vitro: increased endothelial cell branching blocked by mAb D6.1 In vivo: peritoneal carcinosis—increased vessel density (intravital microscopy) abolished by mAb D6	[21]
Highly metastatic rat pancreatic adenocarcinoma BSp73ASML (+/−Tspan8 knockdown)	In vitro: transwell migration and wound healing: reduced in BSp73ASML-Tspan8^kd^ and by mAb D6.1 in BSp73AMSL No effect of D6.1 on BSp73ASML single-cell motility In vivo: delayed metastasis and prolonged survival in BSp73ASML-Tspan8^kd^	[22]
Human colorectal cancer cell lines: Isreco1 and Is-Co029(Tspan8) HT-29, SW480 and SW480-Tspan8	Patients: IHC: Tspan8 high expression correlated with worse prognosis In vitro: single cell-motility on collagen I increased by Ecad, p120ctn and EGFR RNAi when Tspan8 is expressed. This effect is reversed by anti-mouse Tspan8 mAb Ts29.1. No effect of mAb Ts29.1 or Ts29.2 on proliferation In vivo (nude mice): tumor growth reduced by i.p. injection of mAb Ts29.2 No effect on angiogenesis (IHC–CD34 labeling) Tumor growth inhibition by i.v. injection of [^177^Lu]DOTA-Ts29.2	[12,19,23,24]
Human ovarian cell line—effect of Tspan8 RNAi, Tspan8-LEL-Fc, Tspan8-LEL IgG (human Ab selected by phage display)	In vitro: invasion in Matrigel-coated Transwell is inhibited by the 3 reagents In vivo: partial metastasis inhibition (SK-OV3-Luc) by i.v. injection of Tspan8-LEL IgG	[25]

i.v.: intravenous, i.p.: intraperitoneal, i.f.p: intrafootpad and s.c.: subcutaneous.

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
