# Peer review of "Targeting the Tetraspanins with Monoclonal Antibodies in Oncology: Focus on Tspan8/Co-029"

_cancers, 2019, doi:10.3390/cancers11020179_

Round 1
Reviewer 1 Report
This is a very interesting and useful review about targeting tetraspanins by antibodies. The authors focus their review in immunotherapy and radioimmunotherapy approaches.
I recommend that this short review should be accepted after minor revisions as follow:
- disulfide should not be written disulphide or di-sulphide, be correct this in text and Figures
- line 104: antibody-drug conjugates instead of antibody conjugates
- line 123: rephrase "reviewed elsewhere" by "previously reviewed" to avoid repetition
- point 3. ubiquitous antigens can be targeted by antibodies using the pro-body approcah developed by CYTOMX: this could be discussed
- lines 164, 174, 196, 232, 233: in vivo, in vitro and et al. should be in italic
- lines 279 and 282: 77Lu] should be replaced by [77Lu]
- line 282: "using 100 more" may be replaced by "using 100 times more" ?
and in references:
- end pages should be given for references 11, 27, 41, 42, 46
- fourth name for reference 17 should be corrected.
Reviewer 2 Report
Wording is awkward in places and should be clarified by a native English speaker, but this short review should be useful for the general Tspan readership.
Reviewer 3 Report
Dear authors,
the present review covers recent findings about the role and potential function of tetraspanins, potential mediators of cancer metastasis. Hence, their targeting with neutralizing antibodies seems a plausible step in cancer therapy. However, I am a bit confused by the data reported in patients (human protein atlas) and tumor-derived cell lines. Hence prior publication authors should address the following questions:
1.) How do authors explain the interaction of E-cadherin or EGFR and tetraspanins? Why is the knock-down of E-cadherin associated with increased motility? Initiation of EMT or activation of Wnt/ß-catenin signaling?
2.) Are there comparable data pointing out a role of tetraspanins in melanoma, breast cancer or lung cancer or glioblastoma? Is the role of tetraspanins in mesenchymal and epithelial tumors comparable?
3.) Are there data of clinical trials using tetraspanine-neutralizing antibodies e.g. in ClinicalTrials.gov? If yes, please add the corresponding identifiers. Are there any other cellular models with knock-down, knock-out or over expression of tetraspanines that might provide some mechanistic insight? How do authors explain that the low expression of e.g. Tspan8 is less favorable than a high level as observed in the human protein database?
4.) Colorectal cancer cell lines like HT29, HCT116, SW480 are long-term established and although they have been derived from tumors these cell lines not necessarily show a close resemblance of the initial tumor. So: can authors please speculate about the different role of tetraspanins in tumors, long-term and short-term established cell lines? The latter ones mostly grow under 3D (organoids) but not 2D culture conditions.
5.) The in vivo data are absolutely convincing! However, how do authors explain these findings in conjunction with patients data? Is a low level or a high level of tetraspanins favorable?
6.) Could authors speculate a bit more about the role of tetraspanins in cancer?
